# Coupled Fredkin and Motzkin chains from quantum six- and nineteen-vertex models

Zhao Zhang[1*] and Israel Klich[2]

**1** SISSA and INFN, Sezione di Trieste, via Bonomea 265, I-34136, Trieste, Italy
**2** Department of Physics, University of Virginia, Charlottesville, VA, USA
* zhao.zhang@su.se

January 3, 2023

## Abstract

We generalize the area-law violating models of Fredkin and Motzkin spin chains into two dimensions by building quantum six- and nineteen-vertex models with correlated interactions. The Hamiltonian is frustration free, and its projectors generate ergodic dynamics within the subspace of height configuration that are non negative. The ground state is a volume- and color-weighted superposition of classical bicolor vertex configurations with non-negative heights in the bulk and zero height on the boundary. The entanglement entropy between subsystems has a phase transition as the $q$-deformation parameter is tuned, which is shown to be robust in the presence of an external field acting on the color degree of freedom. The ground state transitions between area- and volume-law entanglement phases with a critical point where entanglement entropy scales as a function $L \log L$ of the linear system size $L$. Intermediate power law scalings between $L \log L$ and $L^2$ can be achieved with an inhomogeneous deformation parameter that approaches 1 at different rates in the thermodynamic limit.

# 1 Introduction

Entanglement entropy (EE) and its scaling has been a central theme of quantum many-body physics, not only because entanglement is a unique feature in the quantum world by itself, but also for their crucial role in determining the computational complexity of the numerical simulations of quantum many-body systems, indication of topological order and understanding of the holographic principle and black hole entropy. While EE of a generic eigenstate in the Hilbert space is shown to scale with the systems size [1], EE of the ground states of gapped local Hamiltonians are generally observed to obey the so-called area-law, scaling with the size of the boundary. A milestone in the study of area-law has been Hastings' rigorous proof of the result in one-dimensional systems [2]. Recently, a similar result in two-dimension has been proven for frustration-free models [3, 4]. While area-law has been ubiquitous in gapped systems, plenty of examples of area-law violation has also been found in various gapless systems. (1+1)-dimensional critical system described by a conformal field theory has EE of logarithmic scaling [5]. EE of free fermions in dimension $d$ scales as $L^{d-1} \log L$ [6]. On the other hand, beyond logarithmic violations have only been known in one dimension so far.

Quantum vertex and height models are an invaluable tool for the description of phases of strongly correlated systems [7–10]. They often emerge as an efficient description of quantum dimer models where strong local constraints facilitate the existence of a well defined "height" degree of freedom. In such models, a ground state may be well described in terms of a height field and its fluctuations. When coupled to other local degrees of freedom in such a way that the height field remains single valued, it is possible to enrich the model to use the fluctuating height field in order to further mediate correlations. One of the most spectacular examples of such a behavior is exhibited in the colored Motzkin and Fredkin spin chains [11–17], where the height degree of freedom can assist in generating an extensive entanglement entropy in the ground state. Such ground states thus exhibit a maximal violation of entanglement "area law". It is important to note that the height field, due to continuity constraints, cannot solely by itself reproduce such behavior [18]. In this paper we construct a bicolor six- and nineteen-vertex models that admit exactly such behavior, in analogy with the recent lozenge tiling based model we have presented [19]. Our models are frustration free, with ground states being superpositions of surfaces with colorings, when viewed along a horizontal or vertical direction, obeying the coloring rules of arrays of colored Fredkin or Motzkin spin chains.

The paper is organized as follows. In Sec. 2, we first introduce the six-vertex construction of coupled Fredkin chains, with the Hamiltonian and its ground state explicitly written. In Sec. 3, the EE scaling of the ground state is extracted from a field theory description of the random surfaces in the ground state superposition, showing an entanglement phase transition of the $q$-deformation parameter. Sec. 4 sketches a similar nineteen-vertex construction of coupled Motzkin chains with based on the previous sections. Finally, a summary and discussions of future direction is given in Sec. 5.

# 2 Six-vertex construction of coupled Fredkin chains

The degrees of freedom of the model live on the edges or bonds between vertices of a square lattice of linear size $L$. They can be decomposed into two arrays of one-dimensional spin-$\frac{1}{2}$ chains, one horizontally and one vertically aligned. Each of the edges in the array of horizontal (resp. vertical) chains can have a spin $S^{\text{h}}$ (resp. $S^{\text{v}}$) either up or down (resp. left or right) corresponding to $\pm\frac{1}{2}$. These two sets of degrees of freedom are coupled to each other by the

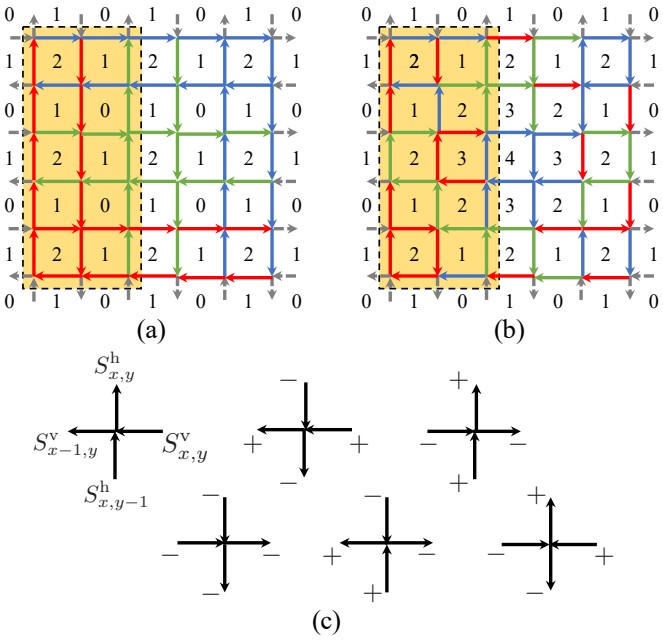

Figure 1: (a) A coloring of the minimal height configuration configuration of a lattice of linear size 6. Three colors are used to manifest the matching pattern. The shaded area with dashed line boundary marks the subsystem A. Then numbers in the plaquettes indicate the height configuration in the dual lattice. (b) A coloring of the maximal height configuration. The heights in the i'th square of plaquettes counting from boundary inward alternates between $i$ and $i+1$.(c) The convention of positive direction of the spins living on the array of horizontal chains $S^{\mathrm{h}}$ and vertical chains $S^{\mathrm{v}}$ at vertex $(i,j)$, along with the other 5 allowed vertex configurations.

six-vertex rule in Fig. 1 (c), enforced by the bulk local Hamiltonian

$$H_0 = \sum_{x,y=2}^{L-1} (S^{\mathrm{h}}_{x,y} - S^{\mathrm{h}}_{x,y+1} - S^{\mathrm{v}}_{x,y} - S^{\mathrm{v}}_{x+1,y})^2. \tag{1}$$

The global Hilbert space can be constrained to the subspace of six-vertex configurations by making the coefficient of this term $V_0 \gg 1$. The boundary spins can be fixed by the Hamiltonian

$$H_\partial = \sum_{y=1}^{L} \left( S^{\mathrm{h}}_{L,y} - S^{\mathrm{h}}_{1,y} \right) + \sum_{x=2}^{L-1} (-1)^x \left( S^{\mathrm{h}}_{x,1} + S^{\mathrm{h}}_{x,L-1} \right)$$
$$\sum_{x=1}^{L} \left( S^{\mathrm{v}}_{x,L} - S^{\mathrm{v}}_{x,1} \right) + \sum_{y=2}^{L-1} (-1)^y \left( S^{\mathrm{v}}_{1,y} + S^{\mathrm{v}}_{L-1,y} \right) + 4L - 4, \tag{2}$$

such that the boundary configurations in Fig. 1 has the minimal energy of 0, and any other configurations will be penalized in proportion to the number of local differences from them along the boundary.

The six-vertex condition allows a well-defined height function $\phi_{x+\frac{1}{2},y+\frac{1}{2}}$ living on the dual lattice of plaquette centers satisfying the rules according to the convention in Fig. 1 (c)

$$\phi_{x+\frac{1}{2},y+\frac{1}{2}} - \phi_{x+\frac{1}{2},y-\frac{1}{2}} = 2S^{\mathrm{v}}_{x,y}, \tag{3}$$

$$\phi_{x+\frac{1}{2},y+\frac{1}{2}} - \phi_{x-\frac{1}{2},y+\frac{1}{2}} = 2S^{\mathrm{h}}_{x,y}, \tag{4}$$

up to a global gauge transformation of shifting the heights by a constant. The effect of boundary Hamiltonian amounts to picking a Dirichlet boundary condition for the ground state wavefunction.

To enrich the entanglement of the ground state, the local Hilbert space of each spin is further enlarged to have either a red of blue color, along with a local Hamiltonian $H_C$ between neighboring up-down spin pairs to match in color

$$H_C = \sum_{x,y=1}^{L-1} \left( |\uparrow_x \downarrow_{x+1}\rangle_y^h \langle\uparrow_x \downarrow_{x+1}| + |\uparrow_x \downarrow_{x+1}\rangle_y^h \langle\uparrow_x \downarrow_{x+1}| + \frac{1}{[2]_r} |\varphi\rangle_{x,y}^h \langle\varphi| + \{h \leftrightarrow v, x \leftrightarrow y\}\right), \quad (5)$$

with $[2]_r := r + r^{-1}$, and

$$|\varphi\rangle_{x,y}^h = r^{-\frac{1}{2}} |\uparrow_x \downarrow_{x+1}\rangle_y^h - r^{\frac{1}{2}} |\uparrow_x \downarrow_{x+1}\rangle_y^h, \quad (6)$$

where the up (resp. down) arrows are used to denote spin $\frac{1}{2}$ in both horizontal and vertical directions. The deformation parameter $r$ plays the role of an external color field, such that when $r = 1$, the ground state will have a uniform superposition of different coloring, while when $r > 1$, the configurations with more red colored spins will be favored.

Since the color Hamiltonian only acts on up-down and left-right pairs, for it to affect all the spins in the system, there must be a net surplus of up (resp. left) spins in any sub-chain counting from left (resp. bottom). In other words, the height function in the dual lattice must stay non-negative and the spins form Dyck paths along the chains in both directions. This can be enforced by the correlated swapping Hamiltonian

$$H_S = \sum_{x,y=2}^{L-1} \sum_{f_h,f_v=\pm} \sum_{c_{1,2}^{h,v}=r,b} \frac{1}{[2]_q} \left| \pi_{x,y,f_h,f_v}^{c_1^h,c_1^v,c_2^h,c_2^v} \right\rangle \left\langle \pi_{x,y,f_h,f_v}^{c_1^h,c_1^v,c_2^h,c_2^v} \right| \quad (7)$$

$$(8)$$

where we have used red, blue, green, orange to represent $c_1^h, c_1^v, c_2^h$, and $c_2^v$ respectively. The total Hamiltonian

$$H = H_0 + H_\partial + H_C + H_S \quad (9)$$

is a frustration-free sum of projection operators, meaning its zero energy ground state is the simultaneous lowest energy eigenstate of each term. Since each term in the Hamiltonian requires a superposition of locally different height and coloring in a particular way, the ground state is therefore a weighted superposition of bicolored six-vertex configurations with alternating heights between 0 and 1 along the boundary, and non-negative heights in the bulk

$$|\text{GS}\rangle = \frac{1}{\sqrt{\mathcal{N}}} \sum_{\phi(\partial P)=\frac{1}{2}} \prod_{x,y=1}^L \theta(\phi_{x,y}) \sum_C{}' r^{\frac{M(C)}{2}} q^{\frac{\mathcal{V}(P)}{2}} |P^C\rangle, \quad (10)$$

where for simplified notation, the height function of each spin is taken to be the average between the heights of its two adjacent plaquettes, the first sum is over all six-vertex configurations $P$ with boundary height $\frac{1}{2}$, the second primed sum is over coloring patterns with spins in the same chain on the same height matching, the volume of a configuration is defined as $\mathcal{V}(P) = \sum_{x,y=1}^{L-1} \phi_{x+\frac{1}{2},y+\frac{1}{2}}$, $M(C)$ is the "warmness" magnetization of coloring $C$, defined as the difference between the number of red and blue spins, and $\mathcal{N}$ is the normalization constant that depends only on $q$ and $r$. The uniqueness of the ground state is guaranteed by the ergodicity of the Hamiltonian (7), which is proven in the appendix.

## 3   Scaling of entanglement entropy

The model has an apparent $D_4$ lattice symmetry, so a cut across the middle along either the horizontal or vertical direction gives the same bipartite entanglement entropy between subsystems. Unlike a quasi-2D model of trivial stacking an array of Fredkin or Motzkin chains, the ground state EE of this coupled 2D model is isotropic. Without loss of generality, we choose a vertical cut as shown in Fig. 1, the ground state (10) can be Schmidt decomposed into a sum over Dyck paths $\vec{\phi}$ and "warmness" magnetization $m(\vec{c})$ of the unmatched colors denoted by $\vec{c}$, which is the difference between the number of red and blue spin pairs among them

$$|\text{GS}\rangle = \sum_{\{\vec{\phi};\vec{c}\}\in\mathcal{D}^2} r^{\frac{m(\vec{c})}{2}} \sqrt{\frac{\mathcal{M}_{\vec{\phi}}^2}{\mathcal{N}}} \left|P_{\vec{\phi}}^{\vec{c}}\right\rangle_{\text{L}} \otimes \left|P_{\vec{\phi}}^{\vec{c}}\right\rangle_{\text{R}}, \tag{11}$$

where $\mathcal{D}^2$ denotes the space of all bicolor Dyck words of length $L$, and

$$\left|P_{\vec{\phi}}^{\vec{c}}\right\rangle_{\text{L(R)}} = \frac{1}{\sqrt{\mathcal{M}_{\vec{\phi}}}} \sum{}' r^{\frac{m_{\text{L(R)}}}{2}} q^{\frac{\mathcal{V}_{\text{L(R)}}(P^C)}{2}} \left|P^C\right\rangle_{\text{L(R)}} \tag{12}$$

are normalized wave functions of the left (resp. right) subsystems, the primed sum is a shorthand notation for summing over "semi-positive definite" six-vertex configurations with height profile $\vec{\phi}$ on the middle boundary and coloring $\vec{c}$ of the dangling spins with their color matched in the other subsystem, and $m_{\text{L(R)}}$ is the redness magnetization of the pairs with color matched within the subsystem, which takes value between 0 and $(L^2 - L - \frac{A(\vec{\phi})+3L}{2} + 1)/2$. $A(\vec{\phi}) = \sum_{x=1}^{L} \phi_y$ is the cross-sectional area of the stepped surface outlined by the height function, as depicted in Fig. 2.

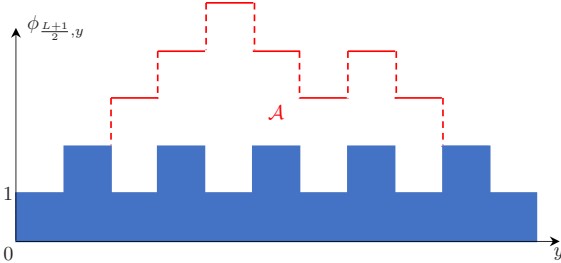

Figure 2: Cross-sectional view of the stepped surface outlined by the height function along the middle cut. Area $\mathcal{A}$ is defined as the sum of heights at each step counting from the minimal heights in Fig. 1 (a).

The normalization constants are given by

$$\mathcal{M}_{\vec{\phi}}^2 = [2]_r^{L^2 - \frac{5L}{2} - \frac{\mathcal{A}(\vec{\phi})}{2} + 1} \sum_{\vec{\phi}(T|_{\text{cut}}) = \vec{\phi}} q^{\mathcal{V}(P)}, \tag{13}$$

and

$$\mathcal{N} = \sum_{\vec{\phi}} [2]_r^{\frac{\mathcal{A}(\vec{\phi}) + 3L}{2} - 1} \mathcal{M}_{\vec{\phi}}^2 \tag{14}$$

$$\equiv [2]_r^{L^2 - L} \sum_{\phi(\partial P) = \frac{1}{2}} q^{\mathcal{V}(P)}. \tag{15}$$

The Schmidt coefficients are given by the probability of height configuration $\vec{\phi}$ with coloring $\vec{c}$ along the cut between subsystems

$$p(\vec{\phi}, \vec{c}) = \frac{r^{m(\vec{c})} \mathcal{M}_{\vec{\phi}}^2}{\mathcal{N}} = p(\vec{c} \mid \vec{\phi}) p(\vec{\phi}), \tag{16}$$

with

$$p(\vec{c} \mid \vec{\phi}) = r^{m(\vec{c})} [2]_r^{-\frac{\mathcal{A}(\vec{\phi})}{2} - \frac{3L}{2} + 1}, \tag{17}$$

can be factorized as a product of probability of having a particular coloring $\vec{c}$ of the unmatched pairs within the subsystems, conditioned on having a Dyck path $\vec{\phi}$ along the cut, and the marginal probability $p(\vec{\phi}) \equiv \sum_{\vec{c}} p(\vec{\phi}, \vec{c})$ of finding such a cross section among uncolored random height configurations. The entanglement entropy decomposed into a piece given in terms of average cross-sectional area of a random height configuration, and another subleading contribution from the fluctuation of the random surface

$$\begin{aligned} S_L(q, r) &= -\sum_{\vec{\phi}, \vec{c}} p(\vec{\phi}, \vec{c}) \log p(\vec{\phi}, \vec{c}) \\ &= \sum_{\vec{\phi}} p(\vec{\phi}) S_L^c(\vec{\phi}, r) + S_L^{\phi}(q) \\ &= \frac{C_r}{2} (\langle \mathcal{A} \rangle + 3L - 2) + S_L^{\phi}(q), \end{aligned} \tag{18}$$

where $S_L^{\phi}(q) = -\sum_{\vec{\phi}} p(\vec{\phi}) \log p(\vec{\phi})$ and in third line we have used

$$S_L^c(\vec{\phi}, r) = -\sum_{\vec{c}} p(\vec{c} \mid \vec{\phi}) \log p(\vec{c} \mid \vec{\phi}) \tag{19}$$

$$= C_r \left( \frac{\mathcal{A}(\vec{\phi})}{2} + \frac{3L}{2} - 1 \right), \tag{20}$$

with

$$C_r = \log[2]_r - \frac{r - r^{-1}}{[2]_r} \log r. \tag{21}$$

This kind of decomposition of entanglement entropy as a result of enlarging the local Hilbert space has also been observed recently in the Bethe Ansatz integrable excited states of a non-integrable one-dimensional multicomponent spin chain [20], which emerges from certain phases of a quasi-2D spin ladder [21].

For any finite $r$, $C_r$ is a finite constant independent of $L$, so the problem is reduced to finding the scaling of the average area $\langle \mathcal{A} \rangle$. That can be done in a field theoretic fashion,

as was previously used to study the dynamics of the one-dimensional Motzkin and Fredkin chains [22,23]. A continuous field of the height configuration can be defined as a piece-wise linear function $\phi(x,y)$, which takes the value of $\phi_{x+1/2,y+1/2}$ on the dual lattice. It is well known that the "entropy" of random surface is captured by a surface tension $\sigma(\nabla\phi(x,y))$ as a function of the height gradient alone [24–27]. Also taking into account the "energy" contribution from volume weighting, we get the partition function

$$Z = \int \mathcal{D}\phi(x,y)e^{\iint dxdy(-\sigma(\nabla\phi(x,y))+(\log q)\phi(x,y))}, \tag{22}$$

where $\mathcal{D}\phi(x,y)$ is a continuous version of

$$\prod_{x,y}\int_0^{+\infty}d\phi_{x+1/2,y+1/2} \equiv \prod_v\int_{-\infty}^{+\infty}dh_v\theta(h_v), \tag{23}$$

and where $\phi$ obeys a Lipschitz condition $|\partial_{x'}\phi,\partial_{y'}\phi| \leq 1$, and $\theta$ is the Heaviside step function.

The linear contribution in $\phi$ is dominant when $q > 1$. To see this explicitly, we substitute

$$x = Lx', \quad y = Ly', \tag{24}$$

which makes

$$\nabla = L^{-1}\nabla', \quad dx = Ldx', \quad dy = Ldy'. \tag{25}$$

The free energy associated to a height configuration then becomes

$$F[\phi] = L^2\iint_0^1 dx'dy'\Big(\sigma\Big(\frac{\nabla'\phi(x',y')}{L}\Big)-(\log q)\phi(x',y')\Big), \tag{26}$$

where $\phi(x',y') = \phi(x,y)$ now satisfy the Lipschitz property of $|\partial_{x'}\phi,\partial_{y'}\phi| \leq L$ instead. The surface tension term counts the entropy of height configurations associated with height variations in a small region with height differences $\partial_{x'}\phi,\partial_{y'}\phi$ on the boundary of the region, and is thus trivially bounded by the entropy density of ice. Thus, in the thermodynamic limit, the surface tension term becomes irrelevant compared to the linear term $(\log q)\phi(x',y')$ when $q \neq 1$. Therefore $F[\phi]$ is minimized by the Lipschitz property for the $q > 1$ case, where minimization of $F$ is achieved with maximal gradient and maximal volume; and by the positivity for the $q < 1$ case, where $F$ is minimized taking $\phi(x,y) = 0$. Therefore, for $q$ larger and smaller than 1 respectively, we have $\langle\mathcal{A}\rangle = O(L^2)$ and $O(L)$. (18) then says $S_L(q,r)$ goes through a phase transition at $q = 1$ from volume scaling to area law.

At the critical point, the height field becomes a massless field conditioned on staying positive. Given that the surface tension is a strictly convex even function of the height variable [24–26, 28–31], the average height was rigorously shown to have the scaling $O(\log L)$, as a result of being repelled by the hard-wall at zero height [32]. This gives the same EE scaling of $O(L\log L)$ as in the recent quantum lozenge tiling model [19], despite the height field of uniform weighted six-vertex model being interacting and not described by a Gaussian free field. This entanglement phase transition can be summarized in the phase diagram in Fig. 3.

The stark phase transition for any $\epsilon = q - 1$ is a consequence of the discontinuity of the partition function $Z$ when the thermodynamic/scaling limit is taken. To obtain an intermediate scaling between $L\log L$ and $L^2$, one can consider a varying $q = e^{\lambda L^{-\alpha}}$ that approaches 1 as $L \to \infty$. Such scaling limits are of interest random surface models, as they admit existence of non-trivial limit shapes [33]. For $\alpha \in (1,2)$, simple scaling argument gives an EE scaling of $L^{3-\alpha}$ with a $\lambda$ dependent coefficient. Whereas for $\alpha \geq 2$, it gives the $L\log L$ scaling, and for $\alpha \leq 1$, it gives the $L^2$ scaling. Interestingly, one can think of this intermediate entropy scaling as the scaling of entropy associated with a square neighbourhood of size $L'$ attached to the corner of a larger lattice where the deformation parameter is inhomogenous, decaying as a function $e^{\lambda d^{-\alpha}}$ of the distance $d$ from the corner of the lattice to the center.

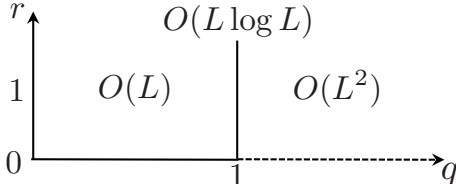

Figure 3: The entanglement phase diagram with two distinct EE scaling separated by the critical line at $q = 1$.

# 4  Nineteen-vertex construction of coupled Motzkin chains

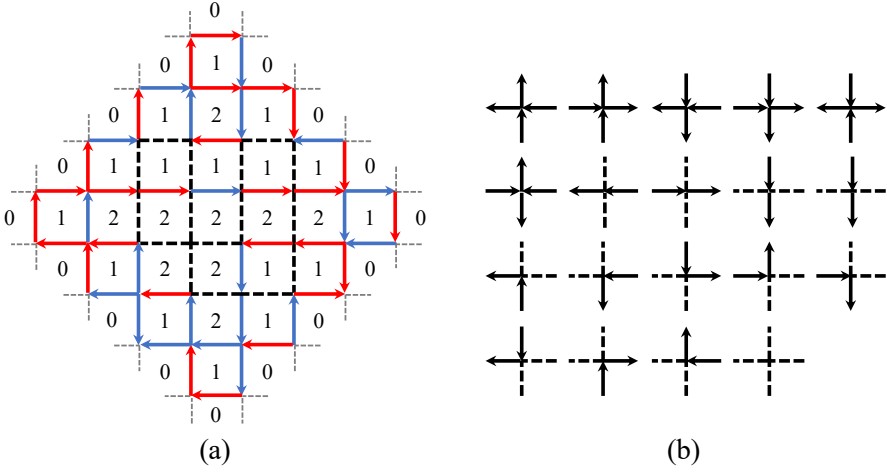

(a)                                                                    (b)

Figure 4: (a) A random coupled Motzkin lattice configuration with Aztec diamond boundary. (b) The 19 vertices in the constrained Hilbert space satisfying eqaul number of in and out arrows.

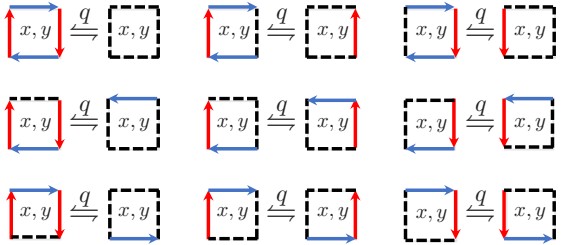

Figure 5: The 9 coupled Motzkin moves that generate the ergodic dynamics in the Hilbert space of random surfaces of height function conditioned to be non-negative.

Building on the previous sections, we introduce a 2D generalization to the Motzkin chain, where each spin takes value of either $\pm 1$ or 0. This can be mapped to solid edges with arrows and dashed edges respectively, giving 19 vertex configurations in Fig. 4 (b) around each of which the height changes by 0. Nineteen-vertex model is a generalization to six-vertex model and is well-studied in the context of classical statistical mechanics [34–40]. Note that classical nineteen-vertex models are mapped to quantum spin-1 chains, by transfer matrix method, which was studied in the context of integrability [41]. Here, we construct a frustration free Hamiltonian by enforcing weighted superposition between pairs of locally different configurations in Fig. 5. The boundary spins in the lattice shown in Fig. 4 (a) is enforced by boundary

Hamiltonians that penalizes −1 spins on the left and bottom side and +1 on the right and top side.

## 5  Conclusions

In this paper we have shown how quantum height models may be enhanced to give a range of exotic entropy scalings. Our models can be viewed as coupled Fredkin and Motzkin chains. They provide another example of a local Hamiltonian with volume scaling of EE. While the height degree of freedom can be described via an appropriate field theory, the addition of the color degrees of freedom within such a description is an interesting open question. Moreover the field theory description only holds for the ground state, while structure of excited states is a subject for additional work.

The equal time correlation functions of the ground state of our model are given by the correlation functions of classical six-vertex model subject to the constraint of positive height. Even in the absence of such a constraint, the analytical result of its two-point correlation functions are only computed for certain boundary conditions such as the domain wall boundary [42–44]. But it's possible to compute them numerically using Markov Chain Monte Carlo method [45]. Adding the non-negative height constraint would pose a challenge to the application of worm or loop-building algorithms, as maintaining the non-negativity would require checking a larger neighborhood as the loops get longer in each update. Another interesting next step in that direction will be the construction of a tensor network characterization for the state, as was done for in the 1D case [46, 47]. Finally, our model in the absence of an internal color degree of freedom is of interest as it promises anomalous slow dynamics and fragmentation analogous to the classical and quantum Fredkin chains in one dimension [48–50].

## Acknowledgements

ZZ thanks Filippo Colomo, Kari Eloranta, Christophe Garban, Hosho Kastura, Yuan Miao, Henrik Røising and Benjamin Walter for fruitful discussions. ZZ acknowledges the kind hospitality of the Galileo Galilei Institute for Theoretical Physics during the workshops "Randomness, Integrability and Universality" and "Machine Learning at GGI".

**Funding information**    The work of IK was supported in part by the NSF grant DMR-1918207.

## A  Ergodicity of the Hamiltonian and uniqueness of ground state

We now show that when the Hamiltonian $H_S$ acts on a properly colored height configuration it generates another such configuration, and that moreover by actions of $H_S$ we can get from any such configuration to any other. Thus the set of non-negative weighted height configurations with Dirichlet boundaries is closed under the operation, with the weighted superposition of states a unique ground state. In complete analogy with the 1D Motzkin and Fredkin chains, starting from a state which violates non-positivity in the bulk, by applying the projectors we create a superposition that will carry the negative region back to the end of the sample to get penalized by the boundary terms. Just as in the Fredkin chain case, in a non-negative height superposition involving a color violation, by reducing the height of unmatched color pairs may be pushed closer until the violation can be detected by local terms.

Let us now check that we can get to the lowest height configuration from any positive height

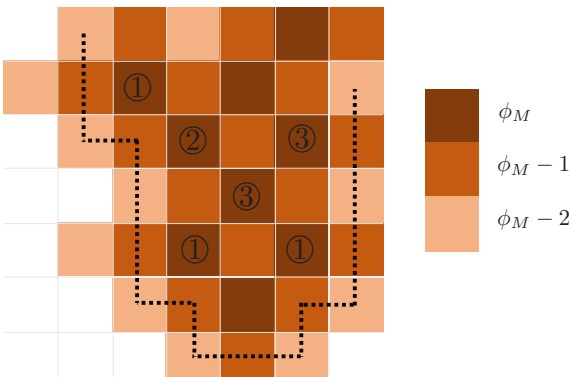

Figure 6: A snippet of plateaux contour of the maximal height $\phi_M$, with the surrounding lower height plaquettes coded by sequentially lighter shades. The heights of the plaquettes numbered ① located at the corners of the contour are ready to be lowered, while those numbered ③ are not. The plaquette ② is an accidental mobile plaquette in this contour configuration despite not lying on the corner.

configuration. Given any six-vertex configuration, there must be a plaquette of maximal height $\phi_M$, which may not necessarily be unique. Their nearest neighbor have height $\phi_M - 1$, but the next-nearest neighbors could either have height $\phi_M$ or $\phi_M - 2$. In the former case, we say the maximal height forms a plateau, while in the latter case, it either lies on the boundary of a plateau, or is isolated. We note that a local maximal height plaquette will have color matched pairs of edges, because of the color rule Eq. (5), therefore it can be removable by one the four moves in (8). Similarly, plaquettes that are on the the boundaries of plateaux are removable if they are at the corner of boundaries (along a straight line of boundary, both sides in the direction of the boundary are not in the right configuration to allow one of the correlated swapping moves), since the next-nearest neighbors are both of the same height. Thus, given any boundary of a plateau, we can always reduce the volume of a surface by first removing the height cubes on the (convex) corners of plateaux boundaries, after which new corners will appear, so that the procedure keeps going. The only scenario such a procedure terminates is when the boundary forms a straight line with the plateau extending to the boundary of the lattice. In that case, both sides of the straight line have the same constant height as the boundary, meaning we have arrived at the lowest height configuration.

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
