# Peer review of "Coupled Fredkin and Motzkin chains from quantum six- and nineteen-vertex models"

_SciPost Physics_

## Round 1 · Referee Report · Yuan Miao (Referee 1) · 2023-2-1

Strengths

1- The authors construct new area-law violating models in 2 dimensions, which has not been reported before.

2- The authors show new type of entanglement transition for the ground state that scales with $L \log L$, which is new.

Weaknesses

1- The inclusion of the colour degree of freedom is not very clear for the non-expert.

2- Though the results about the entanglement entropy scaling are based on rigorous mathematical physics results, the exact derivation (at least for the case without colour degree of freedom) seems possible but missing.

3- The part about 19-vertex model is very brief and it is not obvious to me that the results on the 6-vertex model can be easily generalised.

Report

The article is novel and timely, especially in the field of the area-law violating ground states. It provides new interesting frustration-free Hamiltonian in 2D with ground state entanglement entropy scaling of $\mathcal{O} (L \log L)$. In my opinion, it suffices the criterion of publishing in SciPost Physics.

However, there are few places where the authors could improve, especially for the non-expert readers.

1- The authors should explain better the necessity of the term $H_S$, i.e. eq. (7). It is not obvious to me that this term will enforce spins along both directions to form the Dyck paths. It would be great if the authors could explain more about the motivations to include this term.

2- The authors used some rigorous mathematical results to compute the average height of the vertex model (i.e. entanglement entropy scaling). Of course with the colour of the edges, it is difficult to get exact results other than the scaling. However, from Fig. 3, when $r = 0$ or $1$ (if I'm not mistaken, the ground state should only contain edges with one colour), the entanglement transition seems to be present still, which corresponds to a (classical) 6-vertex model (with equal weights of vertices) with domain-wall boundary conditions. Is it correct to say that the entanglement transition will happen also for the "colourless" case? This is not only an integrable point, as well as a combinatorial point. I'm wondering if it is possible to solve the average height exactly, including the prefactor of the scaling.

3- The generalisation to 19-vertex model case is a bit brief to me. In fact, from Fig. 4, it seems that only edges with arrow have colour, while the dashed lines do not have. The reason is not explained explicitly as far as I see. Moreover, the authors mentioned the connection between the 19-vertex model and integrability. If I'm not mistaken, the authors would like to consider the model while all 19 possible vertices have the same weight. This classical vertex model is not integrable, neither being a special point of the Zamolodchikov-Fateev / Izergin-Korepin ones. I think it would be great if the authors could include a bit more discussions in the quantum 19-vertex model.

4- There are a few typos and more references might be needed to be included, which are listed in the requested changes.

Requested changes

1- In the abstract, the authors wrote "$\textit{The ground state transitions between area- and volume-law entanglement with a critical point}$", which should read "The ground state transitions between area- and volume-law entanglement have a critical point".

2- It would be better if the authors could include a few citations in the first sentence of the introduction, which could provide the non-expert reader some ideas in the field.

3- On page 1, the authors wrote "$\textit{It is important to note that the height field, due to continuity constraints, cannot solely by itself reproduce such behavior}$". I'm wondering if it means that the fluctuation of the height field is important for the violation of the area law in the ground state.

4- At the end of the introduction, the authors wrote "$\textit{Sec. 4 sketches a similar nineteen-vertex construction of coupled Motzkin chains with based on the previous sections. Finally, a summary and discussions of future direction is given in Sec. 5.}$", which should read "Sec. 4 sketches a similar nineteen-vertex construction of coupled Motzkin chains based on the previous sections. Finally, a summary and discussions of future direction are given in Sec. 5."

5- On page 3, Fig. 1, I did not understand the colour green for the edges. Does it mean that those edges can be either red or blue?

6- On page 3, it would be nice to include a few references on the "six-vertex rule", which is called the "ice rule" in the literature. For instance, Lieb's seminal work (Phys. Rev. Lett. 18, 692–694 (1967)) and possibly Baxter's book (Exactly Solved Models in Statistical Mechanics) as general references. The boundary Hamiltonian guarantees the domain-wall boundary condition, where one could cite the a few seminal papers, such as (Commun. Math. Phys 86 (1982), 391), etc.

7- On page 8, caption of Fig. 4, "$\textit{eqaul}$" should read "equal".

8- There are two types of integrable 19-vertex models, corresponding to different underlying quantum symmetries. It seems to me that the authors want to argue that the 19-vertex model considered here is related to the "spin-1 XXZ" (Zamolodchikov-Fateev 19-vertex) model from the papers cited, e.g. [38]. As commented above, I think what the authors considered is not integrable even in the absence of the colour degree of freedom. I suggest the authors to confirm this and probably citing the first few papers such as (Model factorized S Matrix and an integrable Heisenberg chain with spin 1, Yadernaya Fizika 32, 581 (1980)).

  • validity: high
  • significance: high
  • originality: top
  • clarity: good
  • formatting: excellent
  • grammar: excellent

Author:  Zhao Zhang  on 2023-03-27  [id 3517]

(in reply to Report 1 by Yuan Miao on 2023-02-01)

We thank the referee for the positive evaluation of our work and the valuable comments and suggestions. Each individual question or concern raised by the referee has been addressed in the list below.

1- “The authors should explain better the necessity of the term H_S…”

We have changed the notation to labeling the color of individual spins in the involved neighborhood to make the effect of correlated swapping clear.

2- “ The authors used some rigorous mathematical results to compute the average height …”

The referee asked an interesting and important question, although beyond the scope of the current manuscript, which is restricted to the 0<r<1 region as has now been explicitly emphasized in the text. For the colorless (r=0 or 1) case, there cannot be an entanglement phase transition, as the lattice itself serves as a projected entangled pair state (PEPS) description of the ground state, which has entanglement entropy bounded by area law. Nevertheless, the exact enumeration results was employed to compute the entanglement in Ref. 22 by one of us confirming this observation.

3- “The generalization to 19-vertex model case is a bit brief to me…”

We have clarified the difference between our 2D quantum Hamiltonian and the 1D quantum model derived from the classical 2D 19-vertex model, which should be distinguished despite both having spin-1 local Hilbert space. More details of the model are also given now by defining explicitly the part of the Hamiltonian that differs from its six-vertex counterpart.

4- “There are a few typos and more references might be needed to be included, which are listed in the requested changes.”

Requested changes:

1- “In the abstract, the authors wrote…”

We have corrected the typo.

2- “It would be better if the authors could include a few citations…”

We have now added references of the related background.

3- “On page 1, the authors wrote …”

The referee is right about the importance of the fluctuation, however, this sentence is meant to highlight the limitation of fluctuation alone and its inability to generate severe violations of area law. The reasoning is partly addressed in our response to the second point raised by the referee. Nevertheless, we have rephrased the sentence to avoid confusion.

4- “At the end of the introduction, the authors wrote…”

We have corrected the typo.

5- “On page 3, Fig. 1, I did not understand the color green for the edges. Does it mean that those edges can be either red or blue?”

Indeed, green color is used to highlight the pairing of color between up-down spins of the same height, which can be either red or blue, as can every other pair in the figure. It is meant just as a representative coloring of 2^{L^2} possibilities. We have now explained that more clearly in the caption.

6- “On page 3, it would be nice to include a few references on the…”

We thank the referee for the suggestion of references and have now added them.

7-” On page 8, caption of Fig. 4,...”

We have corrected the typo.

8- “There are two types of integrable 19-vertex models,”

We confirm that our model is not related to the integrable spin-1 chain, for the reasons that are now reflected in the manuscript, and have added the reference as suggested by the referee.

---

## Round 1 · Referee Report · Anonymous (Referee 2) · 2023-2-15

Strengths

1- The authors construct explicit examples of two-dimensional quantum models that violate an area-law scaling of entanglement entropy by exploiting classical statistical mechanics models.

2- The entanglement entropy of the ground state exhibits various scalings depending on how the deformation parameter $q$ approaches $1$ in the infinite-volume limit.

Weaknesses

1- There are many typos and unclear points that render the manuscript hard to read.

2- The authors do not discuss the properties of excited states.

Report

The authors construct and study two-dimensional frustration-free models in which an area-law scaling of entanglement entropy is violated. The ground state is a weighted superposition of classical configurations in the corresponding classical statistical mechanics model. By exploiting this connection, the authors pin down the parent Hamiltonian and prove the uniqueness of the ground state. Then the entanglement entropy is evaluated in a field theoretic fashion.

Although this field theoretic part does not look completely rigorous, the obtained results seem to be correct and would stimulate further discussions, as there are only a few examples of models violating an area law in the ground state. However, I cannot recommend the publication of the manuscript in its current form. The present manuscript contains so many typos and unclear points that may make it hard to read. I would suggest the authors address the following issues before I can make a decision.

1- Typos in the Hamiltonian I think something is wrong with either the definition of the Hamiltonian or Fig. 1 (c). I wonder if the summand in Eq. (1) should read $(S^{\rm h}{x,y} - S^{\rm h}}-S^{\rm v{x,y}+S^{\rm v})^2$. Also, there must be a typo in the second line of Eq. (2), as there is no sign in front of the first sum.

2- Dirichlet boundary condition On page 4, the authors write, ``The effect of boundary Hamiltonian amounts to picking a Dirichlet boundary condition for the ground state wave function." What does the Dirichlet boundary condition here mean? It would be great if the authors elaborate on this in the revision.

3- The role of $H_C$ The reader unfamiliar with the colored-Fredkin model may find it difficult to see what this term does. I wonder if the authors can graphically illustrate what the ground state of this term looks like using figures.

4- Definition of $H_S$ As the other referee says, the authors should better explain the term $H_S$. I am not sure if Eqs. (7) and (8) define this term precisely. What if $c^{\rm h}_1 = b$ and $c^{\rm v}_1 = r$? The projection corresponding to this case is summed over in Eq. (7), but does not appear in Eq. (8).

5- Eqs. (11) & (12) I tried to unpack the meaning of Eqs. (11) and (12), but falied. What does the ${\vec c}$ mean? It is not clear to me what the ``semi-positive definite" six-vertex configurations mean. Do they mean six-vertex configurations whose height profile $\phi_{x+1/2, y+1/2}$ is non-negative for all $x,y$? I also wonder if the authors can provide the simplest possible example for small $L$ to show the Schmidt decomposition of the ground state explicitly.

6- Eq. (20) I tried but could not follow the derivation of this identity. Since this identity is quite crucial in relating the entanglement entropy with the average area $\langle {\cal A}\rangle$, maybe the authors can write a more detailed derivation.

7- Excited states As in the one-dimensional Fredkin and Motzkin models, the violation of an area law implies the existence of very low-energy states whose energy goes to zero with increasing the system size. Is it possible to prove this by constructing a variational state along the same lines as in the one-dimensional case?

Minor comments: 8- On page 2, the authors write, ``EE of free fermions in dimension $d$ scales as $L \log L$ [6]." Although this is the result obtained by one of the authors, I would like to point out that this statement holds under the assumption that the free-fermion system considered is gapless. I expect that gapped fermion systems just exhibit an area-law entanglement scaling.

9- On page 4, line 5 from the top: either a red of blue color -> either a red or blue color

10- $\theta$ in Eq. (10) I would suggest the authors define $\theta$ below this equation. I know this is defined below Eq. (23). But symbols should be defined when first introduced.

11- The sentence on page 8, ``This can be mapped to solid edges with arrows and dashed edges respectively, giving 19 vertex configurations in Fig. 4 (b) around each of which the height changes by $0$," does not quite make sense. I guess the authors intended to say that the height does not change if a dashed edge is present on the link between the plaquettes.

Requested changes

See my comments in Report.

  • validity: good
  • significance: good
  • originality: high
  • clarity: low
  • formatting: reasonable
  • grammar: good

Author:  Zhao Zhang  on 2023-03-27  [id 3516]

(in reply to Report 2 on 2023-02-15)

We thank the referee for the careful reading of the manuscript and the helpful suggestions. We have corrected and clarified the text on several occasions in the main text, in addition to the addition of a new section dedicated to the scaling of spectral gap in the volume entanglement phase. The response to each individual question or comment is listed below.

1- “Typos in the Hamiltonian..” We fixed the mentioned typos.

2- “Dirichlet boundary condition. On page 4, the authors write, The effect of boundary Hamiltonian amounts to picking a Dirichlet boundary condition for the ground state wave function." What does the Dirichlet boundary condition here mean? It would be great if the authors elaborate on this in the revision.”

We have explained more carefully our choice of global gauge for the height and the boundary condition on the height function.

3- Role of H_C “The reader unfamiliar with the colored-Fredkin model may find it difficult to see what this term does. I wonder if the authors can graphically illustrate what the ground state of this term looks like using figures…”

We have both added a review section at the beginning to properly introduce the 1D models of colored Fredkin and Motzkin chains, and separated more clearly the terms in H_C and explained their respective roles. (The first line is a penalty on color mismatch at up/down peaks while the second is a term responsible for color mixing within the spin configurations participating in the ground state.)

4- Definition of HS

We have corrected the definition by labeling each individual spin with a color variable that can either take the value of red or blue.

5- Eqs. (11) & (12) “I tried to unpack the meaning of Eqs. (11) and (12)...”

They are now Eqs. (19) and (20). Yes, it exactly means the height values are non-negative, and we have replaced it with the explicit definition to avoid confusion. We have also added the Schmidt decomposition for the 1D model in the review section to make the 2D decomposition easier to understand.

6- Eq. (20) “I tried but could not follow the derivation of this identity...”

(Now Eq. (27).) The derivation has now been given in detail.

7- Excited states

We have added a new section generalizing our proof of the one-dimensional model to show the q>1 phase is gapless in the thermodynamic limit.

8- “On page 2, the authors write, EE of free fermions in dimension…” We have clarified the statement following the referee’s suggestion. 9- “On page 4, line 5 from the top:...”

We have corrected the typo.

10- θ in Eq. (10)

(Now Eq. (18).) We have defined it in the first appearance following the referee’s suggestion.

11- “The sentence on page 8…”

We meant that the 19-vertex constraint defines a U(1) gauge invariance, but we have rephrased the sentence to avoid confusion.

---

## Editorial Decision

resubmitted